# SBMLTOODEJAX: Efficient Simulation and Optimization of Biological Network Models in JAX

**Mayalen Etcheverry**\*
Flowers Team
Inria, Univ. Bordeaux (France)
mayalen.etcheverry@inria.fr

**Michael Levin**
Allen Discovery Center
Tufts University (USA)
michael.levin@@tufts.edu

**Clément Moulin-Frier**
Flowers Team
Inria, Univ. Bordeaux (France)
clement.moulin-frier@inria.fr

**Pierre-Yves Oudeyer**
Flowers Team
Inria, Univ. Bordeaux (France)
pierre-yves.oudeyer@inria.fr

## Abstract

Advances in bioengineering and biomedicine demand a deep understanding of the dynamic behavior of biological systems, ranging from protein pathways to complex cellular processes. Biological networks like gene regulatory networks and protein pathways are key drivers of embryogenesis and physiological processes. Comprehending their diverse behaviors is essential for tackling diseases, including cancer, as well as for engineering novel biological constructs. Despite the availability of extensive mathematical models represented in Systems Biology Markup Language (SBML), researchers face significant challenges in exploring the full spectrum of behaviors and optimizing interventions to efficiently shape those behaviors. Existing tools designed for simulation of biological network models are not tailored to facilitate interventions on network dynamics nor to facilitate automated discovery. Leveraging recent developments in machine learning (ML), this paper introduces SBMLTOODEJAX, a lightweight library designed to seamlessly integrate SBML models with ML-supported pipelines, powered by JAX. SBMLTOODEJAX facilitates the reuse and customization of SBML-based models, harnessing JAX's capabilities for efficient parallel simulations and optimization, with the aim to accelerate research in biological network analysis.

## 1 Introduction

Developing methods to explore, predict and control the dynamic behavior of biological systems, from protein pathways to complex cellular processes, is an essential frontier of research for bioengineering and biomedicine [1]. Systems such as gene regulatory networks and protein pathways play pivotal roles in directing embryogenesis, influencing cellular activities, and governing intricate physiological processes. Understanding the range of behaviors that these systems can exhibit is crucial for comprehending and intervening in various disease states, including cancer [2–4], and for the development of novel bioengineered constructs in synthetic morphology contexts [5–7].

Thus, significant effort has gone in computational inference and mathematical modeling of biological systems, which has resulted in the development of large collections of publicly-available models, typically stored and exchanged on online platforms (such as the BioModels Database [8, 9]) using

---

\*Source code, documentation and online tutorials can be found at `http://developmentalsystems.org/sbmltoodejax`. Please contact Mayalen Etcheverry for any additional questions.

the Systems Biology Markup Language (SBML), a standard format for representing mathematical models of biological systems [10, 11].

Yet, despite the wealth of available SBML models, scientists still lack an in-depth understanding of the range of possible behaviors that these models can exhibit under different initial data and environmental stimuli, and lack effective methods to reveal and optimize those behaviors via external interventions. Except for a set of simple networks where system behavior and response to stimuli can be well understood analytically (or with exhaustive enumeration methods), onerous sampling of the parameter space and time-consuming numerical simulations are needed. This remains a major roadblock for progress in biological network analysis.

On the other hand, recent progress in machine learning (ML) has led to the development of novel computational tools that leverage high-performance computation, parallel execution and differentiable programming and that promise to accelerate research across multiple areas of science, including biological network analysis [12] and applications in drug discovery and molecular medicine [13, 14].

However, to our knowledge, there is no software tool that allows seamless integration of existing mathematical models of cellular molecular pathways (SBML files constructed by biologists) with ML-supported pipelines and programming frameworks. Whereas there exists many software tools for manipulation and numerical simulation of SBML models, they typically rely either on specialized simulation platforms limiting the flexibility for customization and scripting (such as COPASI [15, 16], Virtual Cell [17, 18] and Cell Designer [19, 20]) or provide scripting interfaces in Python or Matlab but rely on backend engines that do not support hardware acceleration or automatic differentiation (like Tellurium [21, 22] and SBMLtoODEpy [23] Python packages, or the Systems Biology Format Converter (SBFC) which generates MATLAB and OCTAVE code [24]).

In this paper we present SBMLTOODEJAX, a lightweight library that seeks to bridge that gap by bringing SBML simulation to the JAX ecosystem, a thriving community of JAX libraries that aim to accelerate research in machine learning and beyond, with diverse applications spanning molecular dynamics [25], protein engineering [26], quantum physics [27], cosmology [28], ocean modeling [29], photovoltaic research [30], acoustic simulations [31] and fluid dynamics [32].

SBMLTOODEJAX aims to integrate this ecosystem and provide tools to accelerate research in biological network analysis. In this paper, we explain how SBMLTOODEJAX facilitates the re-use of existing biological network models, as well as their manipulation in Python projects and AI pipelines, while tailoring them to take advantage of JAX main features for efficient and parallel computations.

## 2   SBMLTOODEJAX

SBMLTOODEJAX is a lightweight library that allows to automatically parse and convert SBML models into Python models written end-to-end in JAX, a high-performance numerical computing library with automatic differentiation capabilities [33]. SBMLTOODEJAX is targeted at researchers that aim to incorporate SBML-specified ordinary differential equation (ODE) models into their pPthon projects and machine learning pipelines, in order to perform efficient numerical simulation and optimization with only a few lines of code.

Taking advantage of JAX's core transformation features, one can easily boost the speed of ODE models time-course simulations and perform efficient search and optimization by running simulations in parallel and/or using automatic differentiation to find derivatives.

SBMLTOODEJAX extends SBMLtoODEpy, a Python library developed in 2019 for converting SBML files into Python files written in Numpy/Scipy [23]. The chosen conventions for the generated variables and modules are slightly different from the standard SBML conventions (and from the conventions used in the original SBMLtoODEpy package) with the aim to accommodate for more flexible manipulations while preserving JAX-like functional programming style.

We refer to the documentation available at https://developmentalsystems.org/ sbmltoodejax/ for additional details on SBMLTOODEJAX's design principles, chosen conventions for the generated variables and modules, and full API docs. Various hands-on tutorial notebooks for loading and simulating biomodels, running parallel executions and performing gradient descent optimization with the generated ODE models are also provided.

# 3 Why use SBMLtoODEjax?

**Simplicity and extensibility**   SBMLtoODEjax retains the simplicity of the SBMLtoODEpy library to facilitate incorporation of the ODE models into one's own Python projects. As shown in Figure 1, with only a few lines of Python code, one can load and simulate existing SBML files. Moreover, one can easily refactor the models to its need.

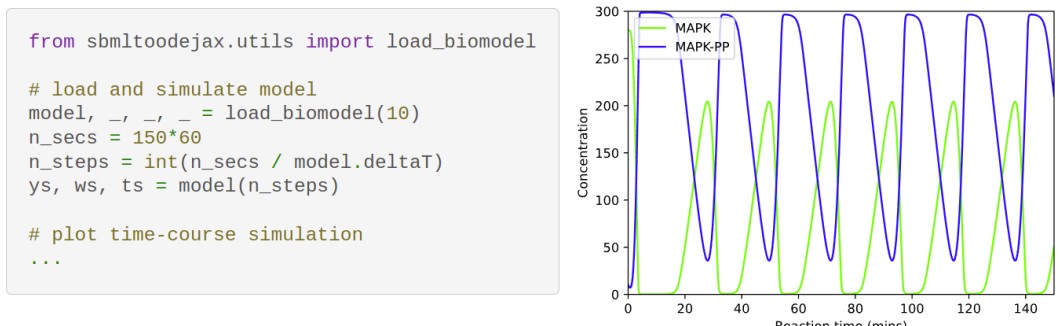

Figure 1: Example code (left) and output snapshot (right) reproducing original simulation results of Kholodenko 2000's paper [34] hosted on BioModels.

**JAX-friendly**   The generated Python models are tailored to take advantage of JAX main features. Model rollouts use `jit` transformation and `scan` primitive to reduce compilation and execution time of the recursive ODE integration steps, which is particularly useful when running large numbers of steps (long reaction times). Models also inherit from the Equinox module abstraction [35] and are registered as PyTree containers, which facilitates the application of JAX core transformations to any SBMLtoODEjax object.

**Efficiency simulation and optimization**   The application of JAX core transformations, such as just-in-time compilation (`jit`), automatic vectorization (`vmap`) and automatic differentiation (`grad`), to the generated models make it very easy (and seamless) to efficiently run simulations in parallel. For instance, as shown in Figure 2, with only a few lines of Python code one can vectorize calls to model rollout and perform batched computations, which is especially efficient for large batch sizes.

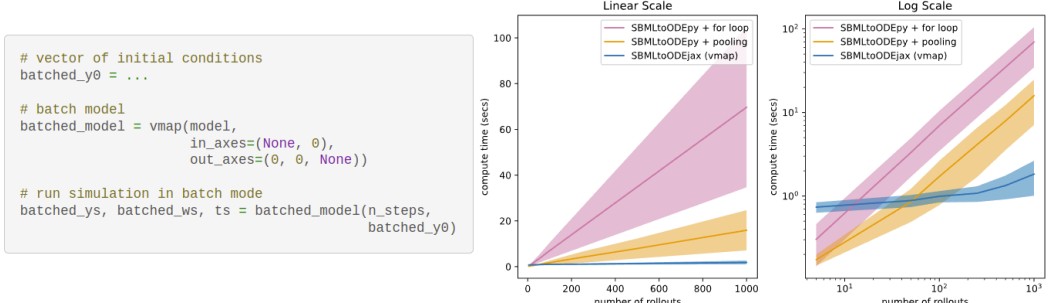

Figure 2: (left) Example code to vectorize calls to model rollout (right) Results of a (rudimentary) benchmark comparing the average simulation time of models implemented with SBMLtoODEpy versus SBMLtoODEjax (for different number of rollouts i.e. batch size). Mean and standard-deviation results of the benchmark, run on 5 models, are displayed. For additional details on the comparison, please refer to our Benchmarking notebook.

As shown in Figure 3, SBMLtoODEjax models can also be integrated within Optax pipelines, a gradient processing and optimization library for JAX [36], allowing to optimize model parameters and/or external interventions with stochastic gradient descent.

Altogether, the parallel execution capabilities and the differentiability of the generated models opens interesting possibilities to design and optimize intervention strategies.

```python
# Load Model
model, y0, w0, c = load_model(model_idx)

# Optax pipeline
@jit
def loss_fn(c, model):
    """loss function"""
    ys, ws, ts = model(n_steps, y0, w0, c)
    loss = jnp.sqrt(jnp.square(ys[y_indexes["Ca_Cyt"]]
                              -target_Ca_Cyt).sum())
    return loss

@jit
def make_step(c, model, opt_state):
    """update function"""
    loss, grads = value_and_grad(loss_fn)(c, model)
    updates, opt_state = optim.update(grads, opt_state)
    c = optax.apply_updates(c, updates)
    return loss, c, opt_state

n_optim_steps = 1000
optim = optax.adam(1e-3)
opt_state = optim.init(c)
train_loss = []
for optim_step_idx in range(n_optim_steps):
    loss, c, opt_state = make_step(c, model, opt_state)
    train_loss.append(loss)

# Plot train loss
plt.plot(train_loss)
plt.ylabel("training loss")
plt.xlabel("train steps")
plt.show()
```

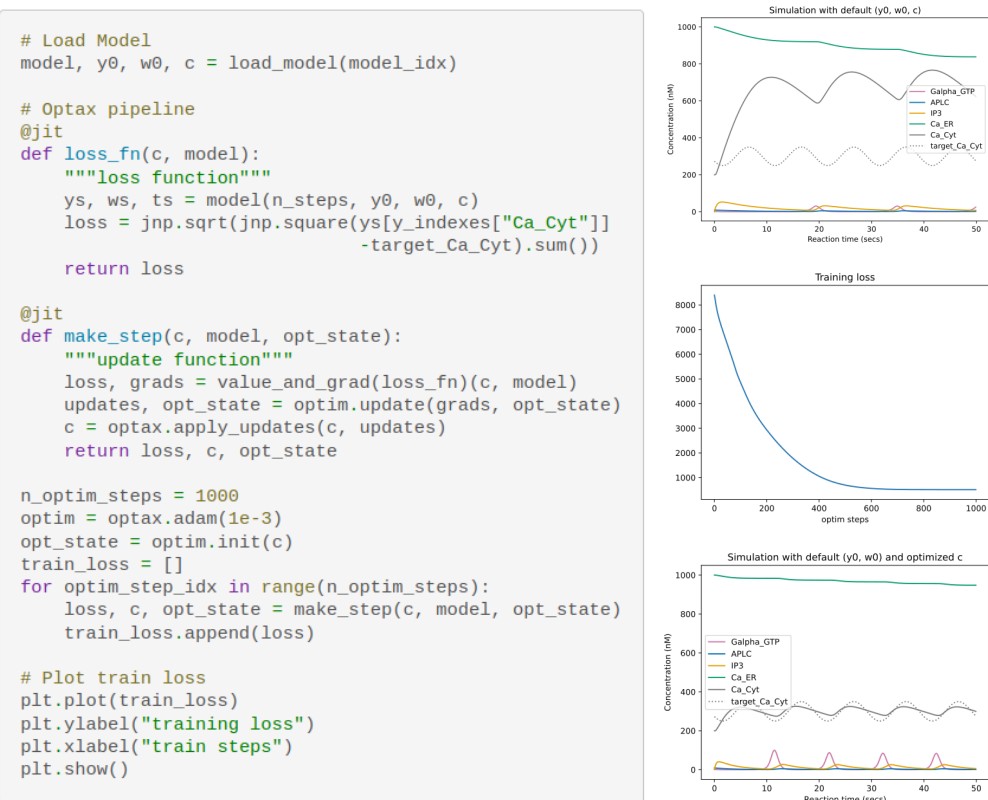

Figure 3: (left) Example of seamless integration with Optax optimization pipeline, with Adam optimizer, over the model kinematic parameters $c$ of the biomodel #145 which models ATP-induced intracellular calcium oscillations. (right-top) Default simulation results of biomodel #145 and (arbitrary) target sine-wave pattern for Ca_Cyt concentration (shown in light gray). (right-middle) Training loss obtained after running the Optax optimization loop. (right-bottom) Simulation results obtained after optimization where we see that the obtained calcium oscillations (dark gray) align with the target one (light gray). The full example is available at our Gradient Descent tutorial.

## 4 Discussion

The development of SBMLTOODEJAX aims to address the need for efficient integration of Systems Biology Markup Language (SBML) models with machine learning frameworks, particularly the JAX ecosystem. This tool simplifies the utilization of SBML-based models in Python projects and machine learning pipelines, and has the potential to be reused in several contexts at the intersection of machine learning and biology.

As an illustration, SBMLTOODEJAX has been used in a recent publication presenting a novel methodology, using diversity search AI algorithms, for exploring the space of possible behaviors of gene regulatory networks, from the perspective of generic problem-solving agents navigating their environment [37]. The authors present several implications that the constructed "behavioral catalogs" could in turn have for biomedicine (control of gene expression via stimuli, not genetic rewiring) and for gene circuit engineering (optimization of network parameters to perform a desired functionality). An interactive version of their paper is available at https://developmentalsystems.org/curious-exploration-of-grn-competencies/.

However, SBMLTOODEJAX is still in its early phase and there are several developments that could be pursued in future work. First, it does not yet handle all possible cases of SBML files, for instance SBML models with events i.e. discrete occurrences that can trigger discontinuous changes in the model. Similarly, SBMLTOODEJAX only integrates one ODE solver for now (`jax.experimental.odeint`), but could benefit from more [38]. Finally, whereas SBMLTOOD-

EJAX is to our knowledge the first software tool enabling gradient backpropagation through the SBML model rollout, applying it in practice can be hard. GRN models are recurrent networks that are generally run for many steps, with each step calling the ODE solver. This can sometimes lead to gradient issues and long backpropagation compute times. Further optimizing the models to be efficiently differentiable might benefit their broader usage.

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
