# OpenReview forum: "SBMLtoODEjax: Efficient Simulation and Optimization of Biological Network Models in JAX"
_NeurIPS.cc/2023/Workshop/AI4Science — NeurIPS2023-AI4Science Poster_

### Official Review · Reviewer_EVDe · 2023-10-22
**Nice tool to support the analysis of SBML models in JAX**

**Rating:** 7
**Confidence:** 3

**Review:**

# Summary

This paper presents the implementation of a JAX package named SBMLtoODEjax that helps to work with biological network models in JAX.
The main aim of the package is to enable users to import networks written in the Systems Biology Markup Language (SBML) and deposited on databases such as the BioModels Database. The package supports resource-efficient optimization and simulation based on imported models.

# Take-home

This tool has potential and should be of interest to a part of the workshop audience.

# Strength
- Supports research advancements by developing tools for efficient data analysis.
- Extended and comprehensible manual and tutorials.
- The Jax implementation greatly accelerates computation time compared to a similar tool implemented in Python.
- Clear and well-written.

# Weaknesses

- The whole ML community does not yet accept JAX, let alone the Bioinformatic one. This may hinder the usage of the package.
- Authors may want to check the SBML2HYB Python package released in 2023, which allows using ML methods on SBML models
> Pinto, José, et al. "SBML2HYB: a Python interface for SBML compatible hybrid modeling." Bioinformatics 39.1 (2023): btad044.
- The documentation contains more details than the paper. If possible, it would be great to expand the paper with more examples for the camera-ready submission.
- In particular, the paper would improve by benchmarking the tool on >1 model (currently only the model from Kholodenko 2000) presenting different characteristics.
- The package does not support importing all SBML models, as stated in the Discussion and detailed in the online documentation. For instance, only 20% of models from the BioModels Database can be imported and used for simulations at the moment. While this may be due to external errors, it can also hinder the usage of the package.
- There is no example of integration of ML pipeline with their tool.

# Minor comments

There are a couple of typos:
- The _P_ in Python language should be capitalized.
- Line 58: “AI pipeline*s*”.
- Lines 49 and 81: SBMLtoODEpy is not capitalized similarly.
- Figures are not well cited: only the number appears but not “Figure” or “Fig.”
- Line 122: “are generally ~~ran~~ *run* for many steps”.
- Line 115: “and ~~they~~ *there* are several developments”

---

### Meta-Review · Area_Chair_fhNu · 2023-10-27

**Recommendation:** Accept (Poster)
**Confidence:** 5

**Metareview:**

This paper proposes a SBML-to-ODE-Jax tool, which can be used to solve many biological phenomena that can be described with an ODE. This is a very promising toolkit, and it would be interesting to see how it can be applied to solve more challenging biological problems.